# Mutation Rate Analysis of RM Y-STRs in Deep-Rooted Multi-Generational Punjabi Pedigrees from Pakistan

**DOI:** 10.3390/genes13081403

**Published:** 2022-08-07

**Authors:** Shahid Nazir, Atif Adnan, Rahat Abdul Rehman, Wedad Saeed Al-Qahtani, Abrar B. Alsaleh, Hussam S. Al-Harthi, Fatmah Ahmed Safhi, Reem Almheiri, Reem Lootah, Afra Alreyami, Imran Almarri, Chuan-Chao Wang, Allah Rakha, Sibte Hadi

**Affiliations:** 1Department of Forensic Sciences, University of Health Sciences, Lahore 54600, Pakistan; 2Department of Anthropology and Ethnology, Institute of Anthropology, School of Sociology and Anthropology, Xiamen University, Xiamen 361005, China; 3Department of Forensic Sciences, College of Criminal Justice, Naïf Arab University of Security Sciences, Riyadh 11452, Saudi Arabia; 4Prince Sultan Military Medical City, Makkah Al Mukarramah Road, Al-Sulimaniyah, Riyadh 12233, Saudi Arabia; 5Department of Biology, College of Science, Princess Nourah bint Abdulrahman University, Riyadh 11671, Saudi Arabia; 6General Department of Forensic Sciences and Criminology, Dubai Police General Headquarters, Dubai 1493, United Arab Emirates

**Keywords:** endogamous, Punjab, Pakistan, deep-rooted pedigrees, RM Y-STRs, mutation rates

## Abstract

Y chromosome short tandem repeat polymorphisms (Y-STRs) are important in many areas of human genetics. Y chromosomal STRs, being normally utilized in the field of forensics, exhibit low haplotype diversity in consanguineous populations and fail to discriminate among male relatives from the same pedigree. Rapidly mutating Y-STRs (RM Y-STRs) have received much attention in the past decade. These 13 RM Y-STRs have high mutation rates (>10^−2^) and have considerably higher haplotype diversity and discrimination capacity than conventionally used Y-STRs, showing remarkable power when it comes to differentiation in paternal lineages in endogamous populations. Previously, we analyzed two to four generations of 99 pedigrees with 1568 pairs of men covering one to six meioses from all over Pakistan and 216 male relatives from 18 deep-rooted endogamous Sindhi pedigrees covering one to seven meioses. Here, we present 861 pairs of men from 62 endogamous pedigrees covering one to six meioses from the Punjabi population of Punjab, Pakistan. Mutations were frequently observed at DYF399 and DYF403, while no mutation was observed at DYS526a/b. The rate of differentiation ranged from 29.70% (first meiosis) to 80.95% (fifth meiosis), while overall (first to sixth meiosis) differentiation was 59.46%. Combining previously published data with newly generated data, the overall differentiation rate was 38.79% based on 5176 pairs of men related by 1–20 meioses, while Yfiler differentiation was 9.24% based on 3864 pairs. Using father–son pair data from the present and previous studies, we also provide updated RM Y-STR mutation rates.

## 1. Introduction

Y chromosomal microsatellites or short tandem repeats (Y-STRs) play an important role in forensic molecular biology [1,2,3]. Most commonly, Y-STRs are used to determine the male component of DNA mixtures when a high female background is present [4,5] or to rebuild paternal relationships between male individuals [6]. Currently, large and growing reference databases now exist for estimating Y-STR haplotype frequencies among worldwide human populations or ethnic groups (e.g., http://www.yhrd.org (accessed on 18 July 2022) or http://usystrdatabase.org (accessed on 18 July 2022)). Commercially available Y-STR kits (Yfiler, Powerplex 23 and Yfiler plus) are valuable, but there are some limitations to their use in forensic investigations. Overall, the haplotype diversities (HD) of these sets of Y-STRs are suitable for outbreeding populations, usually at 0.995 and higher [7,8]. However, the ability to discriminate between individuals is lower than that of the autosomal STRs. In endogamous populations or populations that recently ran into size contraction followed by quick expansion [9], or have particular cultural practices such as patrilocality [10,11,12], the currently used Y-STR panels provide limited resolution due to the overall reduced Y chromosome diversity.

The major drawback of currently used Y-STRs is that they are unable to exclude close or distant patrilineal relatives of the suspect from having deposited the biological material instead of the suspect himself. These STRs may be helpful in cases where close or distant male relatives may be involved because of their relatively low mutation rates of only a few mutations per thousand generations per locus [6,13,14,15,16,17].

In 2010, Ballantyne et al. [15] reported mutation rates of 186 Y-STRs in 2000 DNA-confirmed father–son pairs. This study identified 13 Y-STR markers with markedly higher mutation rates of about 10^−2^ and termed them rapidly mutating (RM) Y-STRs.

In the current study, we have, for the first time, used this newly developed kit in multigenerational deep-rooted pedigrees, and provided empirical evidence of the ability of the 13 RM Y-STRs to improve paternal lineage resolution by analyzing 861 pairs of men from 62 endogamous pedigrees covering the first to sixth meioses from the Punjabi ethnic group of Punjab, Pakistan, which has not been investigated with rapidly mutating Y-STRs. More than 22% of these pairs cover fifth and sixth meiosis, which is unique and has not been analyzed with rapidly mutating STRs before.

## 2. Materials and Methods

### 2.1. Pedigree Samples and DNA Extraction

Initially, 62 pedigrees of 1–4 generations consisting of 861 pairs of men were located in different areas of Punjab (Attock, DG Khan, Faisalabad, Kalabagh, Lahore, Mianwali, Rawalpindi and Sargodha), Pakistan. A total of 327 blood samples were collected from 63 family members. Pedigree of these family members was generated according to their oral records (1 to 4 generations), which were later on confirmed by National Identity Card (NIC). In case of any confusion, a family registration certificate (FRC) was applied using NADRA (National Database and Registration Authority) online paid services. All participants gave their informed consent either orally (in case they could not write) or in writing after the study aims and procedures were carefully explained to them. This study was approved by the ethical review board of the University of Health Sciences Lahore Pakistan and was in accordance with the principles of the Declaration of Helsinki made at the 64th WMA General Assembly, Fortaleza, Brazil, in October 2013. All blood samples were stored at −20 °C before DNA extraction. DNA was isolated from blood using ReliaPrep™ Blood gDNA Miniprep System (Promega, Madison, WI, USA) according to the manufacturer’s instructions. The quantities of extracted DNA samples were determined using a NanoDrop spectrophotometer (Thermo Scientific, Wilmington, DE, USA), and the final concentration of DNA was diluted to 1 to 2 ng/μL.

### 2.2. PCR Amplification and Genotyping

PCR co-amplification of 13 rapidly mutating Y-STR loci (DYF387S1, DYF399S1, DYF403S1ab, DYF404S1, DYS449, DYS518, DYS526ab, DYS547, DYS570, DYS576, DYS612, DYS626 and DYS627) was performed in a 5-dye fluorescence-based multiplex reaction using the RM-Yplex assay [18]. First, 1–2 ng of the target DNA was amplified according to the conditions described elsewhere [18]. Thermal cycling was conducted under the following conditions: 95 °C for 10 min; 20 cycles of 94 °C for 30 s, 55 °C for 45 s, 72 °C for 60 s; and a final extension of 72 °C for 45 min. All loci were amplified in a GeneAmp PCR System 9700 thermal cycler (Applied Biosystems, Foster City, CA, USA). To further confirm the father–son pair relationships (non-paternity events), the AmpFLSTR™ Identifiler™ PCR Amplification Kit was used according to the manufacturer’s recommendations. Amplified products were analyzed regarding GS600 LIZ size standard and Allelic Ladder using an ABI 3500 genetic analyzer (Applied Biosystems, Foster City, CA, USA) with the POP-6TM polymer (Life Technologies, Carlsbad, CA, USA). Samples were analyzed using GeneMapper^®^ID-X software version 1.2, (Waltham, MA, USA) at a threshold of 50 RFU.

### 2.3. Confirmation of Mutations

Father–son pairs which showed mutations were sequenced using the primers and PCR conditions described elsewhere [18]. The PCR reaction was carried out in a 20 µL volume containing 2 µL of 10 × LA PCR™ Buffer II (Mg^2+^ Plus) (TaKaRa Bio Inc., Dalian, China), 2 µL dNTP Mixture (2.5 mM each) (TaKaRa), 0.25 µL TaKaRa LA Taq™ (5 units/μL) (TaKaRa), 14.75 µL ddH2O and 1 µL of genomic DNA. The PCR products were purified by centrifugation by using a Supreme-02 tube (Takara) and were then sequenced directly with primers. The sequencing reaction was carried out in 20 µL reaction volume, which contains 1 µL of PCR product, 8 µL of BigDye (2.5×), 1 µL of each primer (F and R) (3.2 mM) and 10 µL of dH2O. Then, purified PCR products were used for sequencing by capillary electrophoresis using 3500 Genetic Analyzer (Thermo Fisher Scientific, Waltham, MA, USA) according to the manufacturer’s manual.

### 2.4. Statistical Analysis

The rate of differentiation among male relative pairs was calculated as the number of differentiated pairs of relatives by one or more Y-STRs divided by the total number of male relative pairs on that particular meiosis or degree of relationship (i.e., pair members separated by 1–20 meioses). Mutation rates were calculated as the number of mutations divided by the number of allele transmissions, and binomial standard deviation was used to calculate the mutation rates’ 95% confidence intervals (CI), which are available via http://statpages.org/confint.html (accessed on 18 July 2022). Haplotype diversities were calculated as
(n/n−1)(1−∑fi2)
where *n* is the number of samples, and *fi* is the frequency of the *i*-th haplotype.

## 3. Results and Discussion

### 3.1. Non-Paternity Issues

A non-paternity event (also known as misattributed paternity, not parent expected, NPE) occurs in genetics when someone presumed to be an individual’s father is not the biological father. Non-paternity events are common in pedigrees, and these events were observed in the Pakistani population dataset [6,16]. To overcome this issue related to non-paternity events, we split pedigrees where we observed non-paternity events. In some pedigrees, only one individual was involved in non-paternity. After measuring the genetic distance, that individual was removed from the pedigree and considered an individual haplotype. Individuals with mutation events happening on three or more than three RM Y-STRs were removed. This threshold was derived from our previous studies [6,14,15,16,19]; none showed mutations at more than three RM Y-STR markers, and later on, these non-paternity events were confirmed with autosomal STRs (AmpFLSTR™ Identifiler™ PCR Amplification Kit) (Appendix A).

### 3.2. Male Relative Differentiation from the Current Study

We analyzed the set of 13 RM Y-STRs which was described previously [6,14,15,16,19,20] in 327 Punjabi males belonging to 62 pedigrees of two to four generations, representing a total of 861 pairs of men related by one to six meioses. Genotyped data at 13 rapidly mutating Y-STRs from these pedigrees are summarized in Appendix A. Among these 861 pairs, 512 (59.46%) were differentiated by at least one of the thirteen RM Y-STR markers (Table 1). More specifically, 29.7% of the 135 pairs at the first meiosis (father/sons pair), 39.76% of the 171 pairs at the second meiosis (brothers, grandfathers/grandsons), 60% of the 155 pairs at the third meiosis (great grandfathers/great grandsons, uncles/nephews), 68.13% of the 204 pairs at the fourth meiosis (cousins, grand-uncles, grand-nephews), 80.95% of the 168 pairs at the fifth meiosis (first cousin once removed) and 78.57% of the 28 pairs at the sixth meiosis (second cousin) were differentiated by at least one of the thirteen RM Y-STR markers (Table 1).

We observed that maximum numbers of pairs were differentiated based on a mutation in multi-copy markers. A total of 512 pairs were differentiated by at least one of the thirteen RM Y-STR markers, and 392 (76.56%) pairs were differentiated by multi-copy markers. More specifically, of the male relative pairs separated by 1 (father/sons), 2 (brothers, grandfathers/grandsons), 3 (great grandfathers/great-grandsons, uncles/nephews), 4 (cousin grand-uncles/grand-nephews), 5 and 6 meioses, the 13 RM Y-STR markers separated 75%, 71.95%, 82.80%, 79.14%, 72.05% and 81.81%, respectively. Overall, 76.56% were differentiated by multi-copy markers. Pairs separated by single-copy markers were only 23.43% (Table 2). We also noted an increase in relative pair differentiation when there is an increased number of meioses. Theoretically, increased differential power of STRs was expected because of a greater chance of mutation events happening at independent meiosis.

### 3.3. Male Relative Differentiation from the Current and Previous Studies

The differentiation rate based on 13 RM Y-STRs was 29.7%, obtained from the 135 pairs (Appendix A), which is fairly consistent when compared to the differentiation rates of 26.9%, 24.3% and 20.40% previously obtained from 2378, 428 and 49 father–son pairs, respectively [6,16,19]. Merging previously published data and newly generated data showed an overall RM Y-STR-based father–son differentiation rate of 26.55% from a total of 2990 pairs (Table 2). In the current study, we provide an update on male relative differentiation beyond father–son pairs (*n* = 726) relative to the previous studies (total *n* = 1460) [6,16,19] based on RM Y-STRs. Combing RM Y-STR data (Table 2), males separated by two, three and four meioses are differentiated by 43.69%, 52.86% and 63.22%, which are based on 801, 507 and 533 pairs, respectively. For males separated by five and more meioses, we added 196 pairs (fifth meiosis = 168, sixth meiosis = 28). These results are consistent with previously available studies [6,16,19] and can be reliable.

More and more male relative data should be genotyped with RM Y-STRs in the future to make these differentiation rates dependable, especially for second-degree male relatives (beyond fourth, fifth and sixth meioses).

We have shown this with the differentiation of father–son pairs from the first to the present study. In the first study by Ballantyne et al. [15], the rate of differentiation for father–son pairs was 70% based on 20 pairs. In a follow-up study by Ballantyne et al. [14], this rate decreased to 49% and the number of father–son pairs was 39. The rate of differentiation in the following Ballantyne et al. study [19] was further decreased to 27%, where the number of pairs was 327. In another follow-up study by Adnan et al. [6], the findings were similar, with a 24% differentiation rate in 428 pairs. In the Rakha et al. study [16], we analyzed 49 pairs with a 20.40% differentiation rate, while a 29.70% differentiation rate was observed with 135 pairs here. In a recent study, Fan et al. [17] analyzed 1015 DNA-confirmed father–son pairs with a 20.99% differentiation rate. In contrast, Yfiler™ differentiated 4.81%, 10.51%, 15.32% and 22.95% of related males by 1–4 meioses based on 2474, 590, 333 and 318 pairs studied so far, respectively (Table 2). Nonetheless, conventional Y-STRs tend to have lower mutation rates [13,21], and 9.24% of all related males from the combined studies were differentiated with Yfiler as opposed to 38.79% with RM Y-STRs.

### 3.4. Mutation Rate Estimates from Father–Son Pairs

We also calculated the mutation rates of rapidly mutating Y-STRs based on 135 father–son pairs extracted from 62 pedigrees and combined these data with previous studies [6,14,15,16,19] where the same rapidly mutating Y-STR panel was implemented in other father–son pairs (Table 3). The mutation rates ranged from 0 (0 to 2.70 × 10^−2^) for DYS526a or DYS526b to 1.778 × 10^−1^ (1.174 × 10^−1^ to 2.529 × 10^−1^) for DYF399S1. The average mutation rate across all 13 RM Y-STR markers was 2.91 × 10^−2^ based on a total number of 59 mutations from 2025 meioses. Moreover, 86.44% of mutations were single step and 13.56% were double step. All the mutations were length variation mutations, further confirmed by sequencing. Overall, 52.54% of mutations were losses of mutation and 47.46% were gains of mutation. Double-step mutations were only found on DYF399, DYF403, DYS518, DYS526 and DYS626. Among double-step mutations, 25% were losses and 75% were gains of mutations. Combining these new data with previously available data, the mutation rates ranged from 1.90 × 10^−3^ (7.0 × 10^−4^ to 4.0 × 10^−3^) for DYS626a to 7.45 × 10^−2^ (6.58 × 10^−2^ to 8.39 × 10^−2^) for DYF399S1. These currently estimated mutation rates are most reliable for the 13 RM Y-STR markers, given that the underlying number of meioses ranged from 2949 (DYS570) to 3327 (DYF387S1) between markers. The average mutation rate across all 13 RM Y-STR markers was 1.84 × 10^−2^ (1.72 × 10^−2^–1.96 × 10^−2^) based on a total number of 878 mutations from 47,731 meioses.

### 3.5. Population Genetic Analysis in the Punjabi Men 

Out of 861 pairs, with 72 unrelated Punjabi men, all individuals carried a unique RM Y-STR haplotype with haplotype diversity (HD) of 1. A previous multicenter global study [19] based on RM Y-STRs reported haplotype diversity of 0.9999985 for almost 12,200 male samples around the globe and continental regions, ranging from 0.99836 to 0.9999988. In that multicenter global study [19], several populations also showed the haplotype diversity of 1. We compared our current results with previously published work on populations from Pakistan with Yfiler and RM Y-STRs. On Yfiler 17 Y-STRs, British Pakistanis [22], Punjabi population [23], Kashmiri population [23], Hazara population [9], Sindhi population [24], Youszai population [25] and Pathan population [26] showed discrimination capacities (DCs) of 99.24%, 87.23%, 68.3%, 76.47%, 86.40%, 71.92% and 73.7%, respectively. The samples studied with RM Y-STRs showed HD values ranging from 1 to 0.9921 for Pathan [19], Brahui [19], Punjabi [19], Sindhi [19], Araein [27], Pakistani [6], Punjabi [20] and Sindhi [20] populations. A multicenter global study [19] based on RM Y-STRs also reported meaningfully lower haplotype diversities and lower unique haplotype proportions in consanguineous ethnic groups than in urban and rural groups [19]. The Pakistani population is generally considered highly consanguineous, and in our previous study, 99 pedigrees were sampled from urban (*n* = 48) and rural (*n* = 51) areas. We did not observe any effect on RM Y-STR diversity, which may be due to the small sample size. In the current study, 62 pedigrees initially (after non-paternity events, which were confirmed with AmpFLSTR™ Identifiler™ PCR Amplification Kit, 78 individuals) were sampled from urban (*n* = 42) and rural (*n* = 36) areas, and again, we did not observe any effect on RM Y-STR diversity.

### 3.6. RM Marker Differentiation per Pair

We calculated the differentiation power of 13 RM Y-STRs in 62 pedigrees. On the first meiosis, 59 pairs were differentiated out of 135 and DYF399S1 differentiated 24 pairs (40.67%), while DYS526 a/b did not differentiate any pair. This trend was also followed in other pairs, where DYF399S1 differentiated most of the pairs while DYS526a/b did not differentiate any of the pairs (Table 4). This pattern was concordant with previous studies [6,14,16,19].

## 4. Conclusions

We analyzed 62 consanguineous pedigrees containing 327 individuals with a classic RM Y-STR panel to differentiate male relatives. Usually, conventional Y-STRs fail to differentiate between paternal relatives from the same pedigree. This RM Y-STR panel has provided an exponentially high level of differentiation between paternal relatives, and across the fourth meiosis, the differentiation rate is >80%. Moreover, these 13 RM Y-STRs give us 29–100% paternal lineage differentiation in most of the populations and 26–85% in consanguineous populations where conventional Y-STRs fail to differentiate or have 4% paternal lineage differentiation power. This high rate of differentiation or individualization using classic RM Y-STRs is a great benefit to the field of forensic investigative genetics. Different RM Y-STRs panels (developed by authors and available in [14,17,18,27,28,29]) where multi-copy STRs such as DYF404 primers were further broken down were proven effective in increasing the discrimination capacity [28]. The results of this study are concordant with previous studies [6,14,15,16,19] and have shown considerably increased discrimination power. We also merged these data with previous data; based on 5176 pairs of men related by 1–20 meioses, the overall differentiation rate was 38.79%, and the Yfiler differentiation was 9.24% based on 3864 pairs. Using father–son pair data from the present and previous studies, we also provide updated RM Y-STR mutation rates. However, further studies should be conducted on Pakistani populations, mainly in comparison with commercial kits (Yfiler, Powerplex Y23 or Yfiler Plus kits) to further improve the mutation rate information. This study contributes to globally expanding databases for the set of 13 RM Y-STRs.

## Figures and Tables

**Table 1 genes-13-01403-t001:** Male relative pair differentiation per each of the 13 RM Y-STR (single copy and multi-copy) markers.

Relationship	Total Number of Pairs	Pairs Separated by One or More RM Y-STRs Marker (%)	Pairs Separated by Multicopy Markers (%)	Pairs Separated by Single-Copy Markers (%)
**Father/Son**	135	40 (29.70%)	30 (75%)	10 (25%)
**Grandfather/Grandson**	67	28 (41.80%)	59 (71.95%)	23 (28.05%)
**Brother/Brother**	104	54 (51.92%)		
**Uncle/Son**	147	87 (59.18%)	77 (82.80%)	16 (17.20%)
**Great Grand Father/Great Grandson**	8	6 (75%)		
**1st Cousin/1st Cousin**	190	129 (67.90%)	110 (79.14%)	29 (20.86%)
**Grand Nephew/Uncle**	14	10 (71.42%)		
**1st Cousin**	168	136 (80.95%)	98 (72.05%)	38 (27.95%)
**2nd Cousin**	28	22 (78.57%)	18 (81.81%)	4 (18.19%)
**Total**	**861**	**512 (59.46%)**	**392 (76.56%)**	**120 (23.43%)**

**Table 2 genes-13-01403-t002:** Combined male relative differentiation rates from the current and previous studies [6,14,15,16,19] for the RM Y-STR set and the Yfiler Y-STR set.

Number of Meioses Separating Relative Pairs	Number of Relative Pairs Analyzed for RM Y-STRs	Number of Male Relative Pairs Separated by One or More of 13 RM Y-STRs (%)	Number of Relative Pairs Analyzed for Yfiler Y-STRs	Number of Males Relative Pairs Separated by One or More of 17 Yfiler Y-STRs (%)
1	2990	794 (26.55%)	2474	119 (4.81%)
2	801	350 (43.69%)	590	62 (10.51%)
3	507	268 (52.86%)	333	51 (15.32%)
4	533	337 (63.22%)	318	73 (22.95%)
5	231	178 (77.05%)	63	23 (36.50%)
6	76	56 (73.68%)	48	18 (37.50%)
7	14	4 (28.57%)	14	1 (7.14%)
8	7	5 (71.43%)	7	1 (14.28%)
9	1	1 (100%)	1	1 (100%)
10	7	6 (85.71%)	7	4 (57.14%)
11	6	6 (100%)	6	3 (50%)
12	0	0	0	0
13	2	2 (100%)	2	1 (50%)
14	0	0	0	0
15	0	0	0	0
16	0	0	0	0
17	0	0	0	0
18	0	0	0	0
19	0	0	0	0
20	1	1(100%)	1	0
** *Total* **	**5176**	**2008 (38.79%)**	** *3864* **	**357 (9.24%)**

**Table 3 genes-13-01403-t003:** RM Y-STR mutation rates were obtained from father–son pairs in the present and previous studies.

	Current Study (Pakistan)			Adnan, et al. 2016 (Pakistan) [6]			Rakha, et al. 2018 (Pakistan) [16]			Combined [6,14,15,20] (Worldwide)		
Locus	No. of Mutations	Samples	Mutation Rates (95% Confidence Interval)	No. of Mutations	Samples	Mutation Rates (95% Confidence Interval)	No. of Mutations	Samples		No. of Mutations	Total Samples	Mutation Rates (95% Confidence Interval)
DYS576	4	135	2.96 × 10^−2^ (8.1 × 10^−3^–7.41 × 10^−2^)	6	428	1.4 × 10^−2^ (5.2 × 10^−3^–3.0 × 10^−2^)	1	49	2.04 × 10^−2^ (5.0 × 10^−4^–1.08 × 10^−1^)	44	3250	1.35 × 10^−2^ (9.9 × 10^−3^–1.81 × 10^−2^)
DYF399S1	24	135	1.778 × 10^−1^ (1.174 × 10^−1^–2.529 × 10^−1^)	27	428	6.3 × 10^−2^ (4.2 × 10^−2^–9.1 × 10^−2^)	6	49	1.22 × 10^−1^ (4.63 × 10^−2^–2.47 × 10^−1^)	247	3317	7.45 × 10^−2^ (6.58 × 10^−2^–8.39 × 10^−2^)
DYF387S1	5	135	3.70 × 10^−2^ (1.21 × 10^−2^–8.43 × 10^−2^)	10	428	2.3 × 10^−2^ (1.2 × 10^−2^–4.2 × 10^−2^)	0	49	0 (0–7.25 × 10^−2^)	47	3327	1.41 × 10^−2^ (1.04 × 10^−2^–1.87 × 10^−2^)
DYS570	2	135	1.48 × 10^−2^ (1.8 × 10^−3^–5.25 × 10^−2^)	4	428	9.4 × 10^−3^ (2.6 × 10^−3^–2.4 × 10^−2^)	0	49	0 (0–7.25 × 10^−2^)	32	2949	1.22 × 10^−2^ (7.40 × 10^−3^–1.53 × 10^−2^)
DYS526a	0	135	0 (0–2.70 × 10^−2^)	0	428	0 (0.0–8.6× 10^−3^)	0	49	0 (0–7.25 × 10^−2^)	6	3239	1.90 × 10^−3^ (7.0 × 10^−4^–4.0 × 10^−3^)
DYS626	2	135	1.48 × 10^−2^ (1.8 × 10^−3^–5.25 × 10^−2^)	3	428	7.0 × 10^−3^ (1.4 × 10^−3^–2.0 × 10^−2^)	0	49	0 (0–7.25 × 10^−2^)	33	3212	1.03 × 10^−2^ (7.1 × 10^−3^–1.44 × 10^−2^)
DYS526b	0	135	0 (0–2.70 × 10^−2^)	6	428	1.4 × 10^−2^ (5.2 × 10^−3^–3.0 × 10^−2^)	0	49	0 (0–7.25 × 10^−2^)	38	3174	1.20 × 10^−2^ (8.50 × 10^−3^–1.64 × 10^−2^)
DYS627	1	135	7.40 × 10^−3^ (2.0 × 10^−4^–4.06 × 10^−2^)	3	428	7.0 × 10^−3^ (1.4 × 10^−3^–2.0 × 10^−2^)	1	49	2.04 × 10^−2^ (5.0 × 10^−4^–1.08 × 10^−1^)	45	3289	1.37 × 10^−2^ (1.00 × 10^−2^–1.83 × 10^−2^)
DYS518	5	135	3.70 × 10^−2^ (1.21 × 10^−2^–8.43 × 10^−2^)	14	428	3.3 × 10^−2^ (1.8 × 10^−2^–5.4 × 10^−2^)	2	49	4.08 × 10^−2^ (5.0 × 10^−3^–1.39 × 10^−1^)	67	3079	2.18 × 10^−2^ (1.69 × 10^−2^–2.76 × 10^−2^)
DYS612	1	135	7.40 × 10^−3^ (2.0 × 10^−4^–4.06 × 10^−2^)	14	428	3.3 × 10^−2^ (1.8 × 10^−2^–5.4 × 10^−2^)	1	49	2.04 × 10^−2^ (5.0 × 10^−4^–1.08 × 10^−1^)	60	3290	1.82 × 10^−2^ (1.39 × 10^−2^–2.34 × 10^−2^)
DYS449	1	135	7.40 × 10^−3^ (2.0 × 10^−4^–4.06 × 10^−2^)	2	428	4.7 × 10^−3^ (6.0 × 10^−4^–1.7 × 10^−2^)	2	49	4.08 × 10^−2^ (5.0 × 10^−3^–1.39 × 10^−1^)	35	3140	1.11 × 10^−2^ (7.80 × 10^−3^–1.55 × 10^−2^)
DYS547	3	135	2.22 × 10^−2^ (4.60 × 10^−3^–6.36 × 10^−2^)	8	428	1.9 × 10^−2^ (8.1 × 10^−3^–3.7 × 10^−2^)	2	49	4.08 × 10^−2^ (5.0 × 10^−3^–1.39 × 10^−1^)	64	3202	2.00 × 10^−2^ (1.54 × 10^−2^–2.54 × 10^−2^)
DYF404S1	3	135	2.22 × 10^−2^ (4.60 × 10^−3^–6.36 × 10^−2^)	5	428	1.2 × 10^−2^ (3.8 × 10^−3^–2.7 × 10^−2^)	0	49	0 (0–7.25 × 10^−2^)	41	3262	1.26 × 10^−2^ (9.0 × 10^−3^–1.70 × 10^−2^)
DYF403S1a	2	135	1.48 × 10^−2^ (1.8 × 10^−3^–5.25 × 10^−2^)	11	428	2.6 × 10^−2^ (1.3 × 10^−2^–4.6 × 10^−2^)	1	49	2.04 × 10^−2^ (5.0 × 10^−4^–1.08 × 10^−1^)	80	3027	2.64 × 10^−2^ (2.10 × 10^−2^–3.28 × 10^−2^)
DYF403S1b	6	135	4.44 × 10^−2^ (1.65 × 10^−2^–9.42 × 10^−2^)	3	428	7.0 × 10^−3^ (1.4 × 10^−3^–2.0 × 10^−2^)	3	49	6.12 10^−2^ ( 1.28 × 10^−2^–1.68 × 10^−1^	39	2974	1.31 × 10^−2^ (9.30 × 10^−3^–1.79 × 10^−2^)
Across All	59	2025	2.91 × 10^−2^ (2.23 × 10^−2^–3.74 × 10^−2^)	116	6420	1.8 × 10^−2^ (1.5 × 10^−2^–2.2 × 10^−2^)	19	735	2.59 × 10^−2^ (1.56 × 10^−2^–4.01 × 10^−2^)	878	47,731	1.84 × 10^−2^ (1.72 × 10^−2^–1.96 × 10^−2^)

**Table 4 genes-13-01403-t004:** Male relative pair combinations differentiation by each of the 13 rapidly mutating Y-STR markers.

Markers	Out of All 135 Father/Son Pairs Separated	Out of All 67 Grandfather/Grandson Pairs Separated	Out of All 104 Brother/Brother Pairs Separated	Out of All 147 Uncle /Son Pairs Separated	Out of All 08, Great G. Father/ G. G. Son Pairs Separated	Out of All 190 1st-Cousin/1st-Cousin Pairs Separated	Out of All 14 Grand, Nephew/Uncle Pairs Separated	Out of All 168 1st Cousin1r/1st, Cousin1r Pairs Separated	Out of All 28 2nd Cousin Pairs Separated
DYS526a/b	0 (0%)	0 (0%)	0 (0%)	0 (0%)	0 (0%)	0 (0%)	0 (0%)	0 (0%)	0 (0%)
DYS612	1 (0.74%)	5 (7.46)	5 (4.80)	4 (2.72)	0 (0%)	18 (9.47)	0 (0%)	14 (8.33)	4 (14.28)
DYF399	24 (17.77%)	12 (17.9)	19 (18.2)	27 (18.3)	1 (1%)	60 (31.5)	4 (28.57%)	48 (28.5)	12 (42.85)
DYS547	3 (2.222%)	2 (2.98)	8 (7.69)	4 (2.72)	0 (0%)	24 (12.6)	2 (14.28%)	19 (11.3)	5 (17.85)
DYF404	3 (2.222%)	4 (5.97)	5 (4.80)	1 (0.68)	0 (0%)	8 (4.21)	0 (0%)	6 (3.57)	2 (7.142)
DYS626	2 (1.481%)	0 (0%)	2 (1.92)	0 (0%)	0 (0%)	0 (0%)	0 (0%)	0 (0%)	0 (0%)
DYF403	8 (5.925%)	7 (10.4)	12 (11.5)	16 (10.8)	0 (0%)	56 (29.4)	0 (0%)	43 (25.5)	13 (46.42)
DYS576	4 (2.962%)	2 (2.98)	3 (2.88)	10 (6.80)	1 (1%)	8 (4.21)	3 (21.42%)	6 (3.57)	2 (7.142)
DYS518	5 (3.703%)	2 (2.98)	2 (1.92)	6 (4.08)	0 (0%)	6 (3.15)	2 (14.28%)	4 (2.38)	2 (7.142)
DYS627	1 (0.740%)	1 (1.49)	1 (0.96)	3 (2.04)	0 (0%)	4 (2.10)	1 (7.14%)	4 (2.38)	1 (3.571)
DYS570	2 (1.481%)	1 (1.49)	1 (0.96)	1 (0.68)	0 (0%)	2 (1.05)	0 (0%)	1 (0.59)	1 (3.571)
DYS449	1 (0.740%)	0 (0%)	1 (0.96)	3 (2.04)	0 (0%)	2 (1.05)	0 (0%)	2 (1.19)	0 (0%)
DYF387	5 (3.703%)	2 (2.98)	1 (0.96)	1 (0.68)	0 (0%)	8 (4.21)	0 (0%)	6 (3.57)	2 (7.142)

## Data Availability

All the necessary data which is used to produce these results, is available in Appendix A.

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
