# Peer review of "Mutation Rate Analysis of RM Y-STRs in Deep-Rooted Multi-Generational Punjabi Pedigrees from Pakistan"

_genes, 2022, doi:10.3390/genes13081403_

Round 1

Reviewer 1 Report

The Authors have completed the manuscript review as requested

Reviewer 2 Report

-

This manuscript is a resubmission of an earlier submission. The following is a list of the peer review reports and author responses from that submission.

Round 1

Reviewer 1 Report

Nazir et all investigated 13 RM Y-STRs using pedigree samples in Pakistan. It helps understand the mutation of RM Y-STRs. However, issues should further be clarified in this study.

1, line 63, line 105, the square number and the unit throughout this manuscript should be checked.

2, Line 74, the author claimed to investigate 63 pedigrees, but in line 216, there were 75 unrelated Punjabi men, the number should be checked throughout this manuscript.

3, in table 2, I do not understand this table, please clarify this table and add references

4, in table 3, the combined mutation rate was from Pakistan population? If not, mutation rate should be from Pakistan population.

5, in section 3.7, please give discussion and the reasons about the speculation. In fact, I do not agree with this speculation.

6, the data of Y-STR genotype should be added to help readers understand this study.

Author Response

Manuscript ID: genes-1713064

Manuscript Title: Mutation rate analysis of RM Y-STRs in deep-rooted multi-generational endogamous Punjabi pedigrees from Pakistan

Comments and Suggestions for Authors

Nazir et all investigated 13 RM Y-STRs using pedigree samples in Pakistan. It helps understand the mutation of RM Y-STRs. However, issues should further be clarified in this study.

1, line 63, line 105, the square number and the unit throughout this manuscript should be checked.

Reply: Revised accordingly

2, Line 74, the author claimed to investigate 62 pedigrees, but in line 216, there were 73 unrelated Punjabi men, the number should be checked throughout this manuscript.

Reply: Initially, we have collected the 62 pedigrees, but after genotyping, non-paternity events were seen and we removed those individuals.   That’s why unrelated individuals become 73 instead of 62.

3, in table 2, I do not understand this table, please clarify this table and add references

Reply: This table is presenting the combined male relative differentiation rates from the current and previous studies for the RM Y-STR set and the Y-filer Y-STR set. References are also added.

4, in table 3, the combined mutation rate was from Pakistan population? If not, mutation rate should be from Pakistan population.

Reply: We have added the mutation rates of Pakistani populations along with combined worldwide population along with references

5, in section 3.7, please give discussion and the reasons about the speculation. In fact, I do not agree with this speculation.

Reply: We have added few lines along with references which are in support of our speculation.

6, the data of Y-STR genotype should be added to help readers understand this study.

Reply: Genotyping data is attached as ESM (Supplementary Table 1).

Reviewer 2 Report

The paper describes a study concerning an application of RMY-STRs to pedigrees from Pakistan . 

It is a experimental interesting paper, that can be use from the community in the evaluation of the rate mutation of RM Y-STR.

 However, I think that it would be necessary that the Authors would clarify the following points.

- pag.3 lines 93 -100: The Authors refer to a “commercial kit” regarding the RM Y-STR amplified citing publication 18. Effectively, this is not a commercial kit ( it is not a company production) but a multiplex proposed by the Authors of publication.so I think the Authors have to refer in the text “in according to….”.

Furthermore the Authors report ( line 97) “according to manufacturer’s protocol”: the PCR condition reported are different from those published in article n. 18 and to have refer not a manufacturer but to Authors of the publication.

This concept of “commercial kits” are also reported at pag. 8 line 272, again, to my knowledge, all these are not “commercial kits” (for me the meaning: they sold by a company) but multiplex published by other Authors.

Finally the Authors have to clarify what they report in the last three lanes of the manuscript (pag. 8 line 280) the sentence: “However, further studies should be conducted on Pakistani populations, mainly in comparison with commercial kits to further improve the mutation rate information.” Respect to which they referred to also in relation with the polymorphism the are included in them.

- pag. 3 lanes 132-137: the sentences reported do not clarify what the Authors explain. So they have to re-written them to re-formulate the concept.

Author Response

Manuscript ID: genes-1713064

Manuscript Title: Mutation rate analysis of RM Y-STRs in deep-rooted multi-generational endogamous Punjabi pedigrees from Pakistan

Comments and Suggestions for Authors

Reviewer 2

The paper describes a study concerning an application of RMY-STRs to pedigrees from Pakistan . 

It is a experimental interesting paper, that can be use from the community in the evaluation of the rate mutation of RM Y-STR.

 However, I think that it would be necessary that the Authors would clarify the following points.

- pag.3 lines 93 -100: The Authors refer to a “commercial kit” regarding the RM Y-STR amplified citing publication 18. Effectively, this is not a commercial kit ( it is not a company production) but a multiplex proposed by the Authors of publication.so I think the Authors have to refer in the text “in according to….”.

Reply: Revised accordingly

Furthermore the Authors report ( line 97) “according to manufacturer’s protocol”: the PCR condition reported are different from those published in article n. 18 and to have refer not a manufacturer but to Authors of the publication.

 Reply: Revised accordingly

This concept of “commercial kits” are also reported at pag. 8 line 272, again, to my knowledge, all these are not “commercial kits” (for me the meaning: they sold by a company) but multiplex published by other Authors.

 Reply: Revised accordingly

Finally the Authors have to clarify what they report in the last three lanes of the manuscript (pag. 8 line 280) the sentence: “However, further studies should be conducted on Pakistani populations, mainly in comparison with commercial kits to further improve the mutation rate information.” Respect to which they referred to also in relation with the polymorphism the are included in them.

 Reply: Revised accordingly

- pag. 3 lanes 132-137: the sentences reported do not clarify what the Authors explain. So they have to re-written them to re-formulate the concept.

Reply: Revised accordingly

Round 2

Reviewer 1 Report

The revision satisfies the requirement.

Author Response

(The authors gave the same response as above.)
